# Odor Emissions Factors for Bitumen-Related Production Sites

Enrico Davoli [1],*, Giancarlo Bianchi [1], Anna Bonura [2], Marzio Invernizzi [3] and Selena Sironi [3]

[1] Department of Environmental Health Science, Istituto di Ricerche Farmacologiche Mario Negri IRCCS, Via Mario Negri 2, 20156 Milano, Italy; giancarlo.bianchi@marionegri.it
[2] Centro Regionale Sistemi di Monitoraggio Emissioni Atmosfera, U.O.C. Attività Produttive, ARPA Lombardia, 20124 Milano, Italy; A.BONURA@arpalombardia.it
[3] Politecnico di Milano, Department of Chemistry, Materials and Chemical Engineering "Giulio Natta", Piazza Leonardo da Vinci 32, 20133 Milan, Italy; marzio.invernizzi@polimi.it (M.I.); selena.sironi@polimi.it (S.S.)
* Correspondence: enrico.davoli@marionegri.it; Tel.: +39-02-390-141

**Featured Application: This work is a prerequisite for the definition of odor emission parameters, to be defined between stakeholders and institutions, to achieve emission parameters that are protective for the environment, health and psychophysical well-being for nearby residents of bitumen-related production sites.**

**Abstract:** Bitumen-related production sites are facing increasing difficulties with nearby residents due to odor emissions. This parameter is still not regulated for these plants and little is known about the emissions that these plants have put into the atmosphere with the technologies available today. In this study, emission data from 47 Italian production plants were collected and analyzed to assess which values could describe the current situation in Italy. The results of the analysis showed that emissions are very variable, with odor concentration values between 200 to 37,000 $ou_E/m^3$, but data have a normal distribution. The mean value of the stack odor concentration was found to be 2424 $ou_E/m^3$. It was also possible to calculate emission factors of the plants, such as odor emission rate (OER), which represents the quantity of odor emitted per unit of time, and is expressed in odor units per second ($ou_E \cdot s^{-1}$) and odor emission factor (OEF) per ton of product, expressed in $ou_E/t$. The values obtained were $7.1 \times 10^4$ $ou_E/s$ and $1.4 \times 10^6$ $ou_E/t$. respectively. These data could provide a starting point for the definition of shared values among various stakeholders for the definition of regional guidelines for the emissions of these plants, in order to adjust available technologies towards emission parameters that are protective of the surrounding environment.

**Keywords:** odor; odor emission factors; bitumen; asphalt production

## 1. Introduction

Like many other industrial activities with odor emissions, bitumen-related production sites, like bituminous conglomerate (BC) and bituminous waterproofing membrane plants, are facing increasing difficulties coexisting with residents in proximity of production facilities due to odor emissions. In fact, odor pollution is increasingly perceived as both a health impact [1–4], due to its potential toxic effects, and as an impact on subjective well-being of bitumen-related production sites by residents living in areas exposed to environmental odors. This exposure, from a physiological point of view, has acute health effects, such as nausea, headache, eye and throat irritation [5], and also causes neurogenic inflammation caused by stimuli of chemical irritant receptors in the upper airways and skin [6]. However, there are also psychological effects in people due to stress. Exposure to environmental stimuli, such as odors, in fact, cause an immediate release of catecholamines, which evoke the first signs of stress response, glucocorticoid hormones that are responsible for the effects [7] of stress adaptation, and they also have negative psychological and behavioral consequences [8].

Production plants use different production technologies to pursue, for new plants, objectives that ensure environmental protection, both from an energy point of view, for example with the use of materials that allow production at lower temperatures (warm mix asphalt), and from the point of view of odor emissions, with more confined plants to control fugitive emissions. The emission of odors occurs, in fact, not only through the stack, where the fumes coming from the hot mixing of a bituminous binder with inert materials are emitted, but also from uncontrolled diffuse emissions, due to the storage and handling operations of bitumen at high temperatures.

Emissions may contain particulate matter, hydrocarbons (VOCs), and hydrogen sulfide ($H_2S$), and the levels of emitted pollutants depend on the quality of the raw bitumen material used, the production site, crude oil being used as raw material, the handling and storage of the bitumen, and the inert material, the amount of reclaimed asphalt pavement in the mix and the mixing process used, for example [9]. The different formations of VOCs, including odorous gases, depend on various parameters, such as asphalt temperature, asphalt oxidation and air humidity, and large variations are observed under different operating parameters [10]. Using an electronic nose, different odor patterns have been identified in bitumen produced with crude oils of different origins [11] and in chemical additives and wax modifiers [12], but no data are available about true sensory evaluation (olfactometry). Additionally, to the best of our knowledge, while few data are available on specific odorant's stack emissions, such as hydrogen sulfide [13], there is only one published report on bituminous conglomerate plants' stack emissions odors [14].

In Italy, guidelines are available on the characterization and authorization of gaseous emissions into the atmosphere from limited specific plants considering odor impacts, but there are no specific guidelines for BC plants [14], which do have odor emissions that enter the environment. Currently, authorization for emissions of BC production plants are generally limited to pollutants in stack emissions, such as particulate matter (PM), total hydrocarbons (THC), and polycyclic aromatic hydrocarbons (PAH) and do not consider odor emissions.

The first step to being able to create guidelines is the need to have data on emissions from industry at a national level, with the technology that is currently in use. This is fundamental in order to have benchmarks on which to base a dialogue with interested parties.

In this work we present the data available from several monitoring campaigns focused on odor emissions from hot mix asphalt (HMA) and bituminous waterproofing membrane plants, which will be used to quantitatively describe this specific industry sector. The ultimate goal is to define emission factors related to odor emissions from HMAs based on these data, including estimating an odor emission rate (OER) and a specific odor emission factor (OEF) [15]. These data could provide a basis for parameters used in the future in view of the possible publication of guidelines on the characterization and mitigation of HMA odor emissions.

## 2. Materials and Methods

### 2.1. Odor Emission Factor

The calculation of emission factors is based on the correlation of the emission potential of a particular industrial activity and a parameter typical of the kind of industrial sector under investigation, called an activity index [16]. In the present case, the selected activity index was the annual gross production, expressed in metric tons. Thus, focusing on odor emission, the *OEF* will be expressed in $ou_E/t$. Using this parameter, is will be possible to approximately estimate the OER of a bitumen-related production plant site in its project phase, using the following equation [17]:

$$OER_{BC} = OEF \cdot AI \cdot (1 - \eta_{abt}) \tag{1}$$

where $OER_{BC}$ is the odor emission rate expressed in $ou_E/y$, *OEF* the odor emission factor, expressed in $ou_E/t$, *AI* the activity index, for the present case expressed in t/y, and $\eta_{abt}$ the efficiency of any abatement systems in the plant.

### 2.2. Collection of Odor Emission Data

Data concerning Italian asphalt plants have been collected from studies carried out from all three of the authors' laboratories. In this dataset, some data come from old studies, and information on the plants was not available, as was another dataset of which the samples were collected by the plant directly, and there was no available information regarding the sampling method. The plant dataset considered asphalt concrete (HMA) and bituminous waterproofing membranes as well. Samples were mainly collected from the stack, during normal operation of the facility, at the isokinetic flow sampling point. Fugitive emissions were collected both from the trucks, during loading operations, in the proximity of hot material conveyor belts, and at hot material tank vents. Some facilities had adopted odor control measures, mainly water vapor columns or the use of deodorized liquids around the facilities. A limited number of plants were equipped with abatement systems with air/oil or oil mist separators, mainly for the treatment of diffuse emissions, before they were emitted through the main stack. In the present study, all the odor concentrations data considered for the OEF calculation were used, considering the values before any odor abatement systems present.

The determination of odor concentration was performed according to EN 13725:2003 [18]. Samples were collected in polyethyleneterephtalate (Nalophan NA™) bags with a depression pump in a vacuum chamber, the "lung method", and analyzed within 30 h of collection by dynamic olfactometry. Gases and panel members, data recording, calculation and reporting were conducted according to the European Standard EN 13725. Samples with high moisture concentrations could be pre-diluted with dry odor-free air to prevent condensation on the bag walls. Odor concentrations ($C_{od}$) were reported as European olfactory units/m$^3$ (ou$_E$/m$^3$).

To estimate the OEF of bitumen-related production sites, odor concentration data are not enough; to estimate the extensive quantity of odor impact, the *OER* is needed. The calculation of OER$_i$, for each considered plant *i*, is made with the equation:

$$OER_i = \overline{C_{od,i}} \cdot Q_{air,i} \qquad (2)$$

where *OER*$_i$ (ou$_E$/h) is the odor emission rate representative for each plant *i*, $C_{od,i}$ is the geometric mean of the measured odor concentration at the emission of each plant (ou$_E$/m$^3$), purged by abatement system if present, and $Q_{air,i}$ is the air flow rate of the emission (m$^3$/h), under standard conditions for dynamic olfactometry (293 °K and 101.3 kPa on a wet basis). Where more than one emission chimney was present in a single production site, the sum of each OER was considered. Due to comparison with annual production data, the days and hours of actual operation are taken into account.

For each production site *i*, the total *OER*$_i$ was divided by the gross production value, obtaining and experimental sample of *OEF*$_i$. Diffuse emissions were neglected from the present assessments.

### 2.3. Data Analysis

Sampled plants did not adopt identical technologies and several factors affect odor emissions, such as raw materials, production temperature, additives, odor control technologies, etc., but all emission data were used (see Supplementary Material, Figures S1 and S2) since they were considered representative of the Italian emission scenario. Different analyses were conducted on the data, both graphical and numerical. Since available data had a large variance, spanning over two orders of magnitude, great attention has been paid to how to evaluate data and a normality test was conducted, following the criteria of the Lilliefort tests [19] (considering also a logarithmic transformation). The OER analysis was conducted on selected sites where complete data, including odor emission rate and air flow, were available (see Supplementary Material, Table S1a,b), with a total of 8 plant and 46 emission stack data points. The study considered 85 different stack odor concentration measurements out of 47 different bitumen-related production sites in the Italian territory.

## 3. Results

### 3.1. Odor Concentration Data

Table 1 reports an overview of data available at the time of manuscript preparation. All raw data are reported in the Supplementary Materials (Table S1a,b). Stack emissions are reported both as total data, comprising replicate samples from the same plant, as well as plant data, where samples taken from the same plant were averaged and considered as a single value. Additionally, fugitive emissions are reported. Here, data from different truck loading operations and tank venting samples are reported.

**Table 1.** Overall view of emissions (data are in $ou_E/m^3$). Global data reports of all available data, comprising fugitive emissions and ambient air values.

|  | Global Data | Total Stack Emissions | Plants with Stack Emissions | Fugitive Emissions | Ambient Air |
|---|---|---|---|---|---|
| N = | 113 | 85 | 47 | 12 | 10 |
| mean | 4625 | 4135 | 4494 | 11,085 | 130 |
| median | 1400 | 1500 | 1800 | 1775 | 125 |
| RSD% | 7203 | 6779 | 10,538 | 146,497 | 569 |
| Min | 60 | 200 | 240 | 964 | 60 |
| Max | 49,000 | 37,000 | 20,000 | 49,000 | 220 |

### 3.2. Stack Emissions

All data were converted by log 10 transformation before a normality test. Due to the nature of odor concentration measurements (UNI EN 13725), the statistical calculations were based on the log-transformed data. Figure S2 reports the log transformed dataset.

### 3.3. Estimation of OER and OEF

The experimental data, collected by us and with all the necessary information available to calculate the emission factors, were derived from 46 emission stacks' data points but was limited to eight plants. Only two of them had odor controls systems, with different technologies. For this reason, it was not possible to individually analyze different odor abatement systems or odor control procedures, but they were calculated as a whole, to describe the reality of the sector. This dataset has been organized in order to obtain OER and OEF data. The data presented here consider only the conveyed odor emission from stacks; possible fugitive and diffuse emission are neglected.

For the plants where collected data were enough, the global OER, from conveyed sources, and the related OEF, were calculated. A general overview of OER results is reported in Table 2. As depicted, the obtained yearly OER data span over several orders of magnitude. In addition, OEF shows a wide range of values, but appears fairly narrow compared to that of OER. Figure 1 reports the obtained OEFs, both on a linear and a logarithmic scale.

**Table 2.** Results of OER calculations.

|  | OER [$ou_E/s$] | OER [$ou_E/y$] |
|---|---|---|
| N = | 8 | 8 |
| Arithmetic mean | $1.3 \times 10^5$ | $1.8 \times 10^{12}$ |
| Geometric mean | $7.1 \times 10^4$ | $3.5 \times 10^{11}$ |
| RSD % | 74% | 42% |
| min | $1.1 \times 10^4$ | $3.1 \times 10^{10}$ |
| max | $5.6 \times 10^5$ | $1.3 \times 10^{13}$ |

In order to assess how OEF data are distributed, a Lilliefors (Kolomorov-Smirnov) normality test was conducted on the dataset [19] using MATLAB R2019b. The test was conducted on the data, as they were, and after their logarithmic transformation.

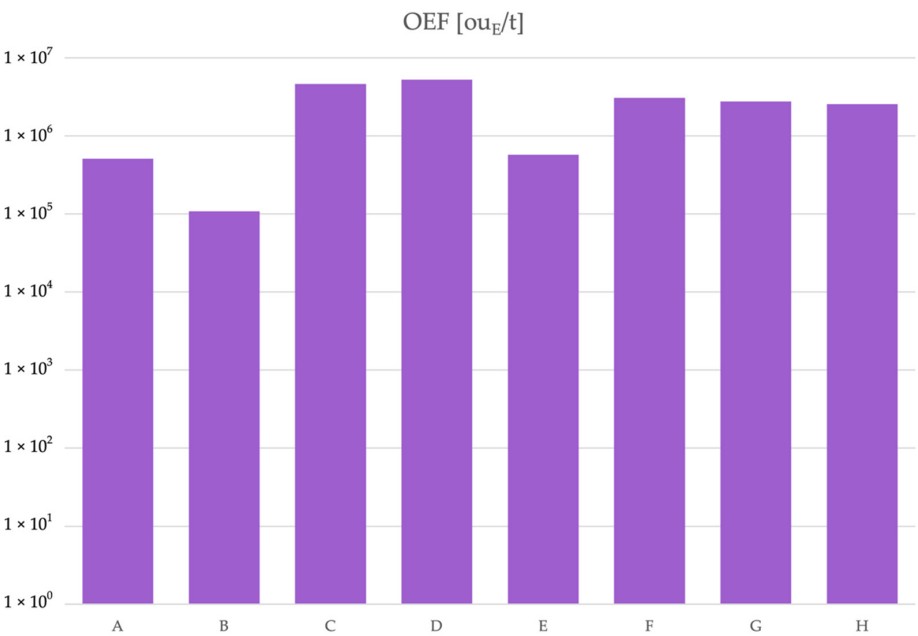

**Figure 1.** OEF data for bitumen-related production sites on a logarithmic scale.

The test on as-is data does not reject the null hypothesis, ($p$-value = 0.38), while it was rejected for the logarithm-transformed data ($p$-value = 0.046). From this information, we can assume that the OEF data show a log-normal distribution [20]. Normal distribution of emissions can be seen in Figure 2, where we report a QQplot of the log-normal distribution of the OEF data. Linearity of all quantiles, close to the diagonal line, with no evident outliers can be observed.

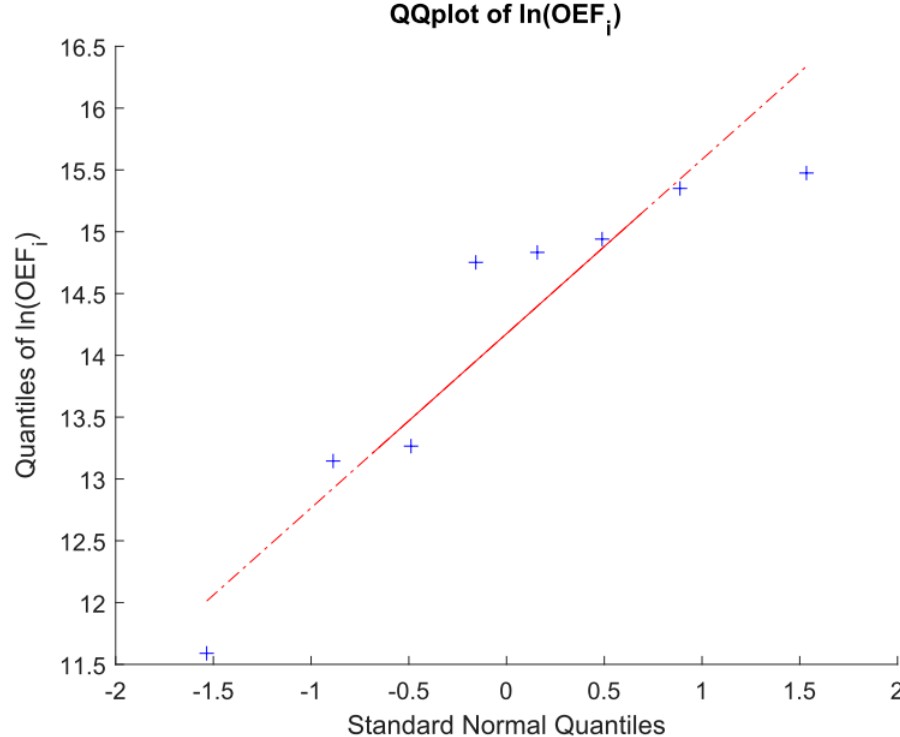

**Figure 2.** QQplot of the logarithm-transformed OEF data. As mentioned in text, due to the nature of odor concentration measurements, the statistical calculations are based on the log-transformed data.

Due to the log-normal distribution of the data, the geometric mean represents the best representation of expected value. Table 3 reports the obtained OEF for the bitumen-related product industry, with its confidence interval (coverage factor k = 1).

**Table 3.** OEF for bitumen-related product industry, with its confidence interval (k = 1).

| OEF [$ou_E$/t] | Lower Level Confidence Interval [$ou_E$/t] | Upper Level Confidence Interval [$ou_E$/t] |
|---|---|---|
| $1.4 \times 10^6$ | $3.7 \times 10^5$ | $5.6 \times 10^6$ |

## 4. Discussion

At present, there are few industrial activities in Europe that have regulated odor emissions. The Joint Research Centre (JRC) published a Reference Document for Waste Treatment where it describes different emission levels and emission factors for different waste treatment activities [21].

In Italy, there is also a lack of well-defined legislation for different industrial activities. In the Testo Unico Ambientale (TUA) (Legislative Decree n. 152 of 2006) odor pollution is generically referred to as air pollution, and there are no specific limits for odor emissions, other than those defined for specific substances for stack emissions. Authorizations of atmospheric emissions are the responsibility of the regions. Regional competences also include the definitions of the best available technologies (BAT), available for the specific treatment of odorous emissions. This is done in order to improve the control of odorous emissions and thus guarantee a higher quality and health of the environment. Those responsible for the assessment and management of air quality are the regions and autonomous provinces that (according to Legislative Decree 351/99) have to carry out an assessment of air quality, either through direct measurements of pollutant levels or through the use of diffusion models. In this context, in Italy, the regions can issue guidelines on odor emissions, while also specifying values for emissions. The Lombardy Region (Giunta Regionale LOMBARDIA: 16 April 2003 n. 7/12764) has published guidelines on the construction and operation of compost production plants, setting emissions to 300 $ou_E$/m$^3$, and the Emilia Romagna Region (Giunta Regionale EMILIA ROMAGNA: 24 October 2011 n.1495) has technical criteria for the mitigation of environmental impacts in the design and management of biogas plants, setting odor emissions to 400 $ou_E$/m$^3$.

Data available from odor stack emissions from bitumen-related production sites plants are very limited and, to the best of our knowledge, are only available for Italian plants [14]. In Italy, guidelines for "Olfactometric characterization and possible mitigation of asphalt plants" [22] have been recently prepared by ARPA, the regional agency for environmental protection, and are now used only for ARPA's internal purposes in the Lombardy Region. In these guidelines two examples of odor emissions are presented, acquired by ARPA technicians, equal to 16,000 $ou_E$/m$^3$ and 1750 $ou_E$/m$^3$ with emission rates of 113,527 and 21,390 $ou_E$/s equal to $1.14 \times 10^5$ and $2.14 \times 10^4$ $ou_E$/s respectively, in line with data reported in this study (see Tables 1 and 2).

The data we present here seem to have a large variability, up to two orders of magnitude, being from 200 to 37,000 $ou_E$/m$^3$. In order to have data that were all-inclusive, all available data were used, even when samples were collected by plant managers. Still, data have a normal distribution. Considering data from 47 plants, and analyzing log-transformed data, the arithmetic mean and median of odor concentration in stack fumes, log($C_{od}$), are similar, 3.38 and 3.36, respectively. The null hypothesis of normality cannot be rejected and data used are normally distributed with a mean value (the inverse logarithm) of $C_{od} = 2424$ $ou_E$/m$^3$. This could be assumed as an immediate starting point for a reference emission value for Italian HMA plants.

To estimate OER and OEF values emission data, production and conveyed air flow data are all necessary [23]. Despite the fact that we have data from 47 plants in Italy, we were only able to use data acquired by us, coming from eight plants, to estimate emission

rates and factors, respectively. OEF is expressed as $ou_E/t$. Values obtained show that odor emission rates are variable, as observed for stack $C_{od}$, but emission factors per metric ton of material produced are more consistent, both for bituminous conglomerate and bituminous waterproofing membrane plants.

## 5. Conclusions

In this paper, a comprehensive set of data on emissions from bitumen-related production sites are collected for the first time. These data allow an overview of the actual odor emission scenario in Italy, with the technologies currently in use. They also make it possible to calculate odor emission factors specific to this type of production. The values calculated in this odor emission study are higher than the values described for waste treatment plants in the Italian guidelines and the mentioned EC document. This could be one reasons why there are increasing complaints from residents in the vicinity of these plants, even though they are not 24-h plants like waste plants. The future definition, possibly with the participation of stakeholders, of reference values could certainly serve both for the planning of new plants in a territory, and also to set reference emission values for existing plants. Emissions could be reasonably achieved with the technologies currently in use in Italy and might be achieved in the future with new technologies with reduced odor emissions.

**Supplementary Materials:** The following are available online at https://www.mdpi.com/article/10.3390/app11083700/s1, Figures S1 and S2: Overall view of raw data acquired. A total of 85 bitumen-related production sites were sampled. Table S1a,b: Complete raw dataset from bitumen related production sites acquired.

**Author Contributions:** Conceptualization, E.D., A.B. and S.S.; methodology, E.D. and S.S.; software, M.I.; formal analysis, G.B. and M.I.; resources, S.S.; data curation, E.D. and M.I.; writing—original draft preparation, E.D.; writing—review and editing, all authors. All authors have read and agreed to the published version of the manuscript.

**Funding:** This research received no external funding.

**Institutional Review Board Statement:** Ethical review and approval were waived for this study, was applicable since the study did not involve humans or animals.

**Informed Consent Statement:** Not applicable.

**Data Availability Statement:** All data used for the study are reported in the Supplementary Materials.

**Conflicts of Interest:** The authors declare no conflict of interest.

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
