# Peer review of "Odor Emissions Factors for Bitumen-Related Production Sites"

_applsci, doi:10.3390/app11083700_

Round 1

Reviewer 1 Report

The manuscript is focused on the quantification of the odor emissions factors processed on the base of emissions data from asphalt production plants in Italy. The objective of the manuscript is good. The following suggestions are needed before the manuscript can be considered for publication.

General remarks

The introduction needs to be supplemented by a more detailed analysis of the current state of knowledge from studies and analyses of other authors.

The measured values of emissions ​​show a large variance of the results from 200 to 37,000. Using statistical processing of data, the value of RSD 74% gives the emission data evaluation a large inaccuracy. The authors should explain and analyze this. In my opinion it is problematic to determine limit values for odor emission on the base of such large variance of the results.  

Detailed comments

Line 116 Add units to the values in equation (2)

Chapter 2.3 The measuring sites/plants, measurements of emissions, and conditions of measurement should be explained, in the text of manuscript the authors state once 47 plants, then 8 plants.

It is appropriate to add a more detailed analysis of factors affecting odor emissions such as production temperature, additives used in production, climatic conditions as temperature and humidity, wind conditions, and/or abatement systems in the plant; which could be decisive in the subsequent statistical processing.

Line 128 Figures S.1 and S.2 are not published in the manuscript

Line 138 Tables S1a and S1b with raw data are not published in the manuscript

Table 1 add units and difference between global data and total data

Line 147 Table S2 is not published in the manuscript

Line 156 Wrong reference

Renumber tables

All tables and figures must be referenced in the text.

Figure 1 shows the same results, one should be deleted according to the authors point out

Figure 2 the authors using log-transformed data in statistical processing and they should explain why used ln-transformation of data in Figure 2

I recommend a deeper discussion of the authors´ results using literature to support the analysis. Lines 183-196 and Lines 218-225 delete or rework, this part deals with local specific decrees, limits, and conditions (region Lombardia in Italy).

224 Error reference

Author Response

The manuscript is focused on the quantification of the odor emissions factors processed on the base of emissions data from asphalt production plants in Italy. The objective of the manuscript is good. The following suggestions are needed before the manuscript can be considered for publication.

We thank the Referee for these comments.

General remarks

The introduction needs to be supplemented by a more detailed analysis of the current state of knowledge from studies and analyses of other authors.

Thank you for this point. It appears that no data is available for odor stack emissions besides our previous report. Several references are found about additives, different oil production sites, temperature etc, but all about materials’ odor, or odorants (like hydrogen sulphide), none on emissions.

A sentence has been added in line 61 and 231.

The measured values of emissions ​​show a large variance of the results from 200 to 37,000. Using statistical processing of data, the value of RSD 74% gives the emission data evaluation a large inaccuracy. The authors should explain and analyze this. In my opinion it is problematic to determine limit values for odor emission on the base of such large variance of the results.  

Again, thank you for raising this point. This was a fact that has been given deep consideration about the statistics. We felt that due to the differences in plants, technologies and materials (bitumen) used, we could describe the national reality as a starting point. We could do this because, although the large variance, the data distribution was normal.

In this respect, paragraph 2.3 has been changed in order to better explain our approach and the results from the statistical analysis. Also in line 185-188 we discussed data distribution.

Detailed comments

Line 116 Add units to the values in equation (2)

Units have been added (lines 126,128 and 129)

Chapter 2.3 The measuring sites/plants, measurements of emissions, and conditions of measurement should be explained, in the text of manuscript the authors state once 47 plants, then 8 plants.

A sentence has been added to explain the site/plants number in line 145-148.

It is appropriate to add a more detailed analysis of factors affecting odor emissions such as production temperature, additives used in production, climatic conditions as temperature and humidity, wind conditions, and/or abatement systems in the plant; which could be decisive in the subsequent statistical processing.

In paragraph 2.3 we added a sentence (line 137-139) to deal with this point.

Line 128 Figures S.1 and S.2 are not published in the manuscript

Line139 has been modified to refer to Supplementary materials 

Line 138 Tables S1a and S1b with raw data are not published in the manuscript

Line146 has been modified to refer to Supplementary materials 

Table 1 add units and difference between global data and total data

Units have been added in the legend of Table 1. Legend and titles have been corrected to better explain global vs. total data.

Line 147 Table S2 is not published in the manuscript

Line153 has been modified to refer to Supplementary materials 

Line 156 Wrong reference

Line 181, the reference has been checked and better presented in the text (Line 180)

Renumber tables

All tables and figures must be referenced in the text.

Figures and tables have been checked. In Line 153 missing tables S1a and S1b have been referenced

Figure 1 shows the same results, one should be deleted according to the authors point out

Thank you, it is correct, we removed the upper figure with linear scale for the reason described in Figure 2 legend (see next point)

Figure 2 the authors using log-transformed data in statistical processing and they should explain why used ln-transformation of data in Figure 2

We changed the legend of Figure 2 to recall (Line 163) and better explain the log-transformation of data.

I recommend a deeper discussion of the authors´ results using literature to support the analysis. Lines 183-196 and Lines 218-225 delete or rework, this part deals with local specific decrees, limits, and conditions (region Lombardia in Italy).

A sentence has been added in the Introduction (Line 65) and in Discussion (Line 231). Also the Discussion Paragraph has been reworded to better explain decrees and limits.

224 Error reference

The reference has been corrected. Hope the problem will not persist in your version (another referee pointed this out).

Reviewer 2 Report

The submitted manuscript presents unique set of data on odor emissions in bitumen-related planets in Italy. The availability of such kind of data is very unique and for this reason, the paper deserves to be published.

I have just two minor comments:

Check missing cross-references (lines 156 and 224).

Line 94: "our laboratories" should be more specified - the authors affiliation is not enough.

Author Response

The submitted manuscript presents unique set of data on odor emissions in bitumen-related planets in Italy. The availability of such kind of data is very unique and for this reason, the paper deserves to be published.

We thank the Referee for this comment

I have just two minor comments:

Check missing cross-references (lines 156 and 224).

References have been corrected. Hope the problem will not persist in your version (another referee pointed this out).

Line 94: "our laboratories" should be more specified - the authors affiliation is not enough.

The sentence has been re-written (Line 99)

Reviewer 3 Report

  1. Section 2.2: more explanation on olfactometry analysis must be presented although it has been metioned that EN 13725 is followed but information on whether panelists are screened based on the EN 13725 or not? how many panellists are used? if outlier data are removed or not? some discussions whether reproducibility and reproductivity tests on data is conducted and how (according to EN 13725)
  2. Table 1: values need units and definition, are the values odour concentration or odour emission rates? From the title of section 3.1, it should be odour concentration but still needs to be mentioned in the table together with the units (OU/m3); tables must stand alone!
  3. Line 156: the sentence “n Error! Reference source not found.” is not clear; please rewrite and clarify the concept.
  4. Line 224 also has the problem of Line 156, I believe it is a matter of using the citation program
  5. Line 171: it must be Table 3 not Table 1
  6. It is better to present more analysis on data for OEF reporting; I mean to present OEF for when odour abatement system/some odour control procedures are available and when they are not, that way the effect of these could be perceived better.
  7. From your available data, please conduct some analysis and present the results on OER on stack emissions and fugitive emissions separately to show their differences and reflect the results in a section, for example, Section 3.1

Author Response

  1. Section 2.2: more explanation on olfactometry analysis must be presented although it has been metioned that EN 13725 is followed but information on whether panelists are screened based on the EN 13725 or not? how many panellists are used? if outlier data are removed or not? some discussions whether reproducibility and reproductivity tests on data is conducted and how (according to EN 13725)

Thank you, this is an important point. Line 117-119 has been added to make explicit the parameters used.

  1. Table 1: values need units and definition, are the values odour concentration or odour emission rates? From the title of section 3.1, it should be odour concentration but still needs to be mentioned in the table together with the units (OU/m3); tables must stand alone!

Thank you again for pointing this out. We changed the legend accordingly.

  1. Line 156: the sentence “n Error! Reference source not found.” is not clear; please rewrite and clarify the concept.
  2. Line 224 also has the problem of Line 156, I believe it is a matter of using the citation program

References have been corrected. Hope the problem will not persist in your version (another referee pointed this out).

  1. Line 171: it must be Table 3 not Table 1

I am deeply sorry but I do not see your point raised here. We checked the Tables and they appear to be correct. In Line 176 we added “OER” to better indicate table 2.

  1. It is better to present more analysis on data for OEF reporting; I mean to present OEF for when odour abatement system/some odour control procedures are available and when they are not, that way the effect of these could be perceived better.

Thank you for this point. We strongly agree, but we had only two plants available with odor abatement/control, and data were not useful for any discussion.

Line 169-170 has been added to better explain this point.

  1. From your available data, please conduct some analysis and present the results on OER on stack emissions and fugitive emissions separately to show their differences and reflect the results in a section, for example, Section 3.1

Again, please see the answer in point 6. Here we have only three plants with fugitive data to make comparisons. We presented all data in supplementary materials, we would like to leave to the readers the possibility to interpret the data or to use it in comparison with their data.

Round 2

Reviewer 1 Report

I have no further comments